# MALLVi: A Multi-Agent Framework for Integrated Generalized Robotics Manipulation

## Abstract

Task-planning for robotic manipulation tasks using large language models (LLMs) is a relatively new phenomenon. Previous approaches have relied on training specialized models, fine-tuning pipeline components, or adapting LLMs with the setup through prompt tuning. However, many of these approaches lack environmental feedback. We introduce the MALLVi Framework, a **M**ulti-**A**gent **L**arge **L**anguage and **Vi**sion framework designed to solve robotic manipulation tasks that leverages closed-loop feedback from the environment. The agents are provided with an instruction in human language, along with an image of the current environment state. After thorough investigation and reasoning, MALLVi generates a series of realizable atomic instructions necessary for a supposed robot manipulator to complete the task. After extracting and executing low-level actions through the downstream agents, a Vision-Language Model (VLM) receives environmental feedback and prompts the framework either to repeat this procedure until success, or to proceed with the next atomic instruction. Our work shows that with careful prompt engineering, the integration of four LLM agents (**Decomposer**, **Localizer**, **Thinker**, and **Reflector**) can autonomously manage all compartments of a manipulation task - namely, initial perception, object localization, reasoning, and high-level planning. Moreover, the addition of a **Descriptor** agent can introduce a visual memory of the initial environment state in the pipeline. Crucially, compared to previous works, the reflecting agent can evaluate the completion or failure of each subtask. We validate our framework through experiments conducted both in simulated environments using VIMABench, RLBench and in real-world settings. Our framework handles diverse tasks, from standard manipulation benchmarks to custom user instructions. Our results show that the agents communicating to plan, execute, and evaluate the tasks iteratively not only lead to generalized performance but also increase average success rate in trials. The essential role of the reflecting in the pipeline is highlighted in experiments.

## 1 Introduction

Natural language tasks are rich, contextual, and often complicated —a simple sentence may be broken down to several smaller subtasks. With the advent of large language models (LLMs), attempts in the task-planning field for robotic manipulation have shifted to using complex language models. The question is clear: "How can we ground abstract instructions into robust, feedback-driven execution in dynamic environments?" As the scope of robotic applications expands, the core challenge has shifted: robots are no longer asked to repeat narrow, pre-programmed motions, but to understand flexible instructions and adapt to unpredictable situations. Existing methods have made progress on this front, typically following two strategies. The first learns behaviors directly from demonstrations, capturing motion trajectories with imitation or policy learning James et al. (2021); James & Davison (2021); Shridhar et al. (2022). The second relies on vision-language models (VLMs) to map natural language and visual input into actions Liu et al. (2024); Kim et al. (2024); Brohan et al. (2023; 2022). Both approaches have been effective in structured settings, but they falter when tasks involve open-vocabulary commands, new objects, or novel environments, where limited semantic understanding and adaptability restrict their use in real-world scenarios Zare et al. (2023); Sapkota et al. (2025).

LLMs offer a promising path forward. They excel at reasoning and problem decomposition, and can translate high-level instructions into structured steps or even executable code Imani et al. (2023); Fang et al. (2024b). Frameworks that harness these capabilities demonstrate that LLMs can serve as powerful planners for robots Liang et al. (2022); Wang et al. (2024a); Fang et al. (2024a); Zawalski et al. (2024). However, these systems often operate in an open-loop manner: they generate plans once, without checking whether execution succeeds in practice. This makes them fragile in dynamic environments, where errors accumulate and hallucinations —plans that look valid in text but fail in the real world— can degrade performance Dhuliawala et al. (2023); Ali et al. (2024); Sun et al. (2024).

Recent research has taken steps toward closing the loop by integrating visual feedback for error detection and replanning Mei et al. (2024); Pchelintsev et al. (2025); Huang et al. (2022); Skreta et al. (2024); Huang et al. (2025). Yet, most of these systems rely on a single, monolithic model, which creates bottlenecks when tasks are ambiguous or when reasoning and perception need to be specialized. Moreover, relying on unconstrained LLMs/VLMs raises safety concerns, as unchecked outputs can lead to unsafe or adversarial behaviors Zhang et al. (2025).

In this paper, we introduce MALLVi (Fig. 1), a multi-agent framework for robotic task planning that directly addresses these challenges. Instead of relying on monolithic models, MALLVi coordinates specialized agents for perception, planning, and reflection, enabling them to collaborate through a shared state. At its core, a decomposer agent translates human prompts into atomic instructions suitable for robotic execution. Subsequent agents handle environmental understanding, object localization, and trajectory planning through a low-level motion planner. A reflector agent continuously monitors the environment via visual feedback, providing a closed-loop that identifies and reactivates only the specific failing agent for efficient error recovery. This distributed, feedback-driven design enables MALLVi to disambiguate instructions, adapt to unexpected changes, and recover from errors—capabilities essential for real-world deployment.

Specifically, we:

- Propose MALLVi, a distributed framework that introduces a genuine multi-agent architecture for robotics, combining LLM-based planning with VLM-based monitoring in a self-correcting process.

- Highlight the novel role of a reflector agent's targeted feedback loop, enabling reflection, error recovery, and adaptation through continuous environmental feedback.

- Validate MALLVi in both simulation (VIMABench Jiang et al. (2022), RLBench James et al. (2020)) and real-world experiments, demonstrating substantial improvements in success rate across diverse manipulation tasks in a zero-shot setting.

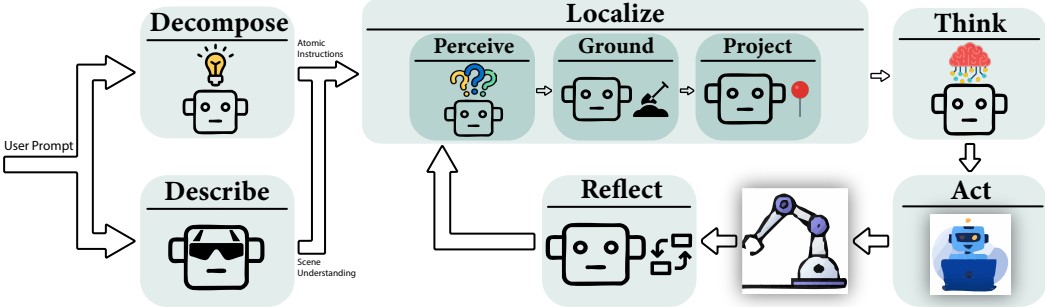

Figure 1: The MALLVi framework architecture. The pipeline processes user prompts through specialized agents: **Decompose** breaks instructions into atomic steps, **Describe** provides scene understanding, **Perceive** processes visual inputs, **Ground** localizes target objects, **Project** generates motion trajectories, **Think** coordinates high-level reasoning, **Act** executes robotic commands, and **Reflect** evaluates outcomes to enable iterative refinement and error recovery.

## 2 RELATED WORK

### 2.1 LLMS AND VLMS FOR ROBOTIC TASK PLANNING

The use of LLMs as high-level planners for robotics has grown rapidly in recent years. Frameworks such as *Code-as-Policies* Liang et al. (2022) treat LLMs as translators that convert natural language into parameterized API calls or executable code. *Inner Monologue* Huang et al. (2022) was an early example of incorporating environmental feedback—including success and failure reports—to guide the LLM's planning. Subsequent work, such as LLM-Planner Song et al. (2023), demonstrated few-shot planning for embodied agents by leveraging LLMs to generate sequences of pre-defined actions. Similarly, *Tree-Planner* Hu et al. (2024) iteratively constructs a task tree using an LLM, decomposing high-level goals into a sequence of executable subtasks.

A central challenge in these approaches is the open-loop nature of the initial plans. To mitigate this, recent research integrates visual feedback into replanning. For example, *Replan* Skreta et al. (2024) combines an LLM for initial planning with a VLM to evaluate execution success, triggering replanning upon failure. *ReplanVLM* Mei et al. (2024) and *LERa* Pchelintsev et al. (2025) similarly use VLMs to detect visual errors and guide corrective action. *Look Before You Leap* Huang et al. (2025) leverages GPT-4V to verify pre- and post-conditions for each planned step, while *CoPAL* Joublin et al. (2024) introduces a self-corrective planning paradigm where the LLM critiques and refines its own actions. Despite these advances, these systems are largely monolithic and limited in modularity, often handling planning, perception, and execution in a tightly coupled manner. They typically lack task-aware decomposition and flexible perception-action pipelines.

### 2.2 MULTI-AGENT COLLABORATION AND REFLECTION

The concept of using multiple LLM agents to collaborate, debate, or critique each other has proven effective for complex problem-solving in non-embodied settings Sprigler et al. (2024); Wang et al. (2024b). This idea is now being applied to robotics. *RoCo* Mandi et al. (2023) is a seminal work that introduces dialectic collaboration between multiple LLM-controlled robots for task planning.

Recent multi-agent approaches have further advanced collaboration in planning and manipulation. *Wonderful Team* Wang et al. (2024c) presents a multi-agent VLLM framework where agents jointly generate action sequences from a visual scene and task description, integrating perception and planning in an end-to-end system. *MALMM* Singh et al. (2024) employs three LLM agents (Planner, Coder, Supervisor) to perform zero-shot block and object manipulation tasks, incorporating real-time feedback and replanning to mitigate hallucinations and adapt to unseen tasks.

Building on prior multi-agent LLM frameworks, our work adopts a modular multi-agent approach for manipulation. Unlike previous methods, MALLVi tightly integrates perception, reasoning, and execution in a collaborative agent pipeline, enabling robust adaptation and closed-loop correction in complex environments.

### 2.3 OPEN-VOCABULARY PERCEPTION

Robust manipulation requires not just object localization, but context-aware grounding to resolve ambiguities (e.g., "the red block" when there are multiple, referring to a past block). While foundational models like OWL-ViT Minderer et al. (2022) and Grounding Dino Liu et al. (2023) provide open-vocabulary detection, they lack situational context. Segmentation models such as the Segment Anything Model (SAM) Kirillov et al. (2023) offer general-purpose, high-quality segmentation masks across diverse object categories. These capabilities make them well-suited for downstream tasks such as grasp point extraction, where precise segmentation underpins reliable interaction. However, they do not incorporate contextual reasoning or task grounding. Approaches such as *SayCan* Ahn et al. (2022) address this gap by incorporating environmental cues, and recent VLMs increasingly combine perception, grounding, and reasoning in a unified framework Bai et al. (2023); Nasiriany et al. (2024). MALLVi builds on these advances by providing a modular, context-aware perception pipeline that integrates detection, segmentation, and grounding to enable precise manipulation.

## 3 METHODOLOGY

The MALLVi framework implements a multi-agent, self-correcting pipeline for robotic manipulation. Given a high-level user instruction and a real-time image of the environment, the system hierarchically decomposes tasks into atomic subtasks, grounds each to visual inputs, plans execution trajectories, and adaptively refines actions based on feedback. Specialized agents communicate through object and memory tags, with automatic retry mechanism ensuring action success.

### 3.1 MULTI-AGENT

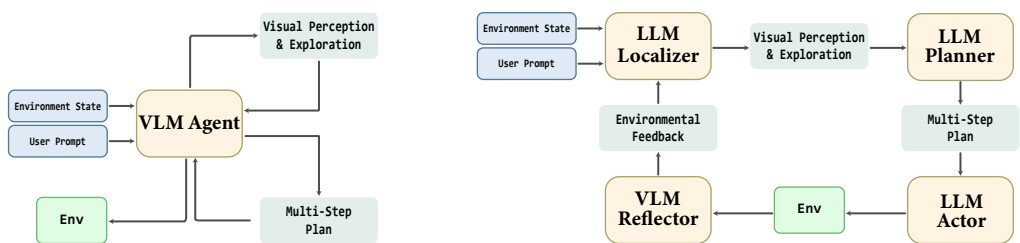

Figure 2: Comparison between single-agent and multi-agent frameworks.

Single-agent frameworks often struggle with maintaining task focus and performing sequential reasoning in complex environments. As illustrated in Fig. 2, MALLVi mitigates these limitations by assigning specialized agents to distinct execution aspects. This modular design reduces hallucinations, supports iterative refinement through closed-loop feedback, and improves overall task execution by maintaining focus and coherence.

#### 3.1.1 DECOMPOSER-DESCRIPTOR AGENTS

The **Decomposer** and **Descriptor** agents operate in parallel at the first stage of the MALLVi pipeline, providing complementary functions for task execution.

The Decomposer agent converts a high-level instruction into a structured sequence of atomic subtasks. Each subtask corresponds to a primitive action in the Actor agent's vocabulary (e.g., move, reach, push) and is annotated with memory tags containing parameters such as object identities, positions, or contextual references. Subtasks are executed sequentially, with a retry mechanism that allows failed steps to be reattempted without replanning the entire sequence. Implementation details, including subtask representation, memory tagging, and fault-tolerant execution, are provided in Appendix section A.2, and A.3.

The Descriptor Agent generates a coarse representation of the environment using a vision-language model (VLM). It identifies objects, extracts the spatial relationships between them, and builds a spatial graph representing the scene. This graph enables the agent to reason about object configurations, constraints, and interactions, providing critical context for downstream perception, grounding, and planning agents.

By running in parallel, the Decomposer focuses on *what* needs to be done (task decomposition), while the Descriptor focuses on *where* and *how* in the environment (scene representation and reasoning). Together, they align task objectives with environmental context from the outset.

#### 3.1.2 LOCALIZER AGENT

- The **Perceptor** agent identifies task-relevant objects from the instruction and labels non-target objects. It refines grasping strategies (as explained in the Projector tool) across multiple attempts, adapting to subtask failures and improving manipulation precision.
- The **Grounder** agent localizes objects in the image plane by integrating outputs from multiple detectors (GroundingDINO and OwlV2) to ensure reliable detection even under partial

failures. Beyond simple fusion, the agent employs a confidence-based selection mechanism: for each object, it weighs predictions from each detector according to model confidence and consistency with the spatial graph provided by the Descriptor agent. This allows the Grounder to provide accurate bounding boxes for downstream planning. By combining multi-model detection and confidence weighting, the Grounder agent ensures robust, high-fidelity localization essential for manipulation in dynamic and unstructured environments.

- The **Projector** tool converts visual perception into actionable 3D grasp points, bridging the gap between scene understanding and robot execution.

  **Grasp Point Extraction:** Leveraging the Segment Anything Model (SAM), the agent identifies candidate grasp points on objects. Object-specific heuristics are applied to select appropriate points (e.g., edges for cylindrical objects, centers for rigid blocks). A verification step ensures that each grasp point lies within the object's segmentation mask, enhancing reliability and precision.

  **3D Projection:** The extracted 2D grasp points are projected into 3D space using the depth map and the pinhole camera model. These 3D coordinates are subsequently converted into joint angles through inverse kinematics, producing executable targets for downstream planning and manipulation agents.

  By integrating grasp point extraction and 3D projection within a single module, the Projector agent provides a direct and reliable interface between visual perception and robotic action. This design enables precise and consistent generation of executable 3D targets, supports closed-loop feedback during task execution, and preserves the modularity of the MALLVi framework, allowing seamless interaction with downstream planning and manipulation agents.

### 3.1.3 THINKER AGENT

The **Thinker** agent is an LLM responsible for translating high-level subtask information into actionable parameters for execution. It retrieves relevant objects (see section 3.1.1) and determines 3D grasp points along with any required rotations.

For tasks without prior memory (memoryless), the Thinker selects pick-and-place positions and rotations directly from the grasp points. For atomic instructions with associated memory tags, the agent identifies either source or target objects using the stored scene representation and spatial relationships, then computes corresponding pick-and-place positions and rotations based on the scene context.

### 3.1.4 ACTOR AGENT

The **Actor** agent executes the subtasks produced by the upstream agent. In both real-world deployments and benchmark scenarios, the Actor interfaces with the environment through a predefined API, receiving the action parameters from the Thinker and performing the corresponding manipulation. This modular design allows the Actor to remain agnostic to high-level reasoning or low-level motion planning while ensuring accurate execution of planned actions.

### 3.1.5 REFLECTOR AGENT

The **Reflector** agent is a VLM responsible for verifying the execution of each subtask in real-time. After the Actor executes a subtask, the Reflector evaluates success using visual feedback. Successfully completed subtasks are removed from the execution queue, while failed subtasks trigger a reattempt from the beginning of the relevant subtask.

We demonstrate that the Reflector agent is essential for executing complex and sophisticated manipulation tasks, as evidenced by our ablation in Tables 1, 2, and 3. This iterative verification mechanism provides robust, closed-loop correction, ensuring that errors are detected and mitigated promptly. By continuously monitoring task execution, the Reflector agent enhances reliability and generalization across diverse and dynamic manipulation scenarios while preserving the modularity of the MALLVi framework.

| High-Level | Demposer Agent |
|---|---|
| | "The main core" |
| **Responsibilities** | |
| • Convert high-level instructions into atomic subtasks. | |
| • Annotate subtasks parameters with tags (object, memory). | |
| • Ensure sequential execution. | |

| High-Level | Descriptor Agent |
|---|---|
| | "Enabling memory utilization" |
| **Responsibilities** | |
| • Build a coarse scene representation from visual input. | |
| • Identify objects and their spatial relationships. | |
| • Provide context for downstream planning. | |

| Mid-Level | Localizer Agent |
|---|---|
| | "The computer vision toolkit" |
| **Responsibilities** | |
| • Localize objects with confidence-based selection. | |
| • Extract candidate source object and target object. | |
| • Project 2D grasp points into 3D coordinates. | |

| Mid-Level | Thinker Agent |
|---|---|
| | "Compilation of compartments" |
| **Responsibilities** | |
| • Compile state info into actionable 3D positions and rotations. | |
| • Compute pick-and-place targets for the task. | |
| • Retrieve objects from memory and scene representation. | |

| Low-Level | Actor Agent |
|---|---|
| | "The doorway to environment" |
| **Responsibilities** | |
| • Receive action parameters from Thinker agent. | |
| • Execute subtasks through a predefined environment interface. | |
| • Remain modular, independent of high-level reasoning. | |

| Mid-Level | Reflector Agent |
|---|---|
| | "Closed-loop controller" |
| **Responsibilities** | |
| • Verify subtask execution in real-time. | |
| • Remove completed subtasks or trigger reattempts on failure. | |
| • Ensure closed-loop error correction for robust manipulation. | |

Figure 3: Analysis of specialized agents and their roles in a multi-agent system. Each agent functions at a designated level (high, mid, or low) to address specific components of task execution, including instruction decomposition, memory utilization, object localization, task reasoning, action execution, and closed-loop feedback provision.

## 4 EXPERIMENTS AND RESULTS

We evaluate MALLVi on both real-world manipulation tasks and benchmarked scenarios from VIMABench and RLBench. These tasks were selected for their alignment with real-world deployment settings, where the agent receives natural language instructions from users and perceives the environment solely through streaming camera input.

**Real-world tasks:** These are designed to reflect common robotic manipulation objectives:

- **Place Food** – tests accurate object placement.
- **Put Shape** – evaluates shape-specific placement.
- **Stack Blocks** – measures precision in stacking.
- **Shopping List** – requires sequential task execution.
- **Put in Mug** – tests fine-grained placement.
- **Math Ops** – evaluates math reasoning.
- **Stack Cups** – tests repetitive stacking skills.
- **Rearrange Objects** – requires organizing multiple objects according to instructions.

Examples of task stages are shown in Fig. 5.

**VIMABench tasks:** We selected a subset of VIMABench partitions with clear real-world analogues:

- **Simple Manipulation** – evaluates basic object handling.
- **Novel Concepts** – tests the agent's ability to generalize to unseen object–instruction combinations.
- **Visual Reasoning** – requires reasoning under perceptual restrictions.
- **Visual Goal Reaching** – measures scene understanding and goal-directed planning.

**RLBench tasks:** These tasks require diverse skill sets in simulated environments:

| Task Name | Prompt | Environment |
|---|---|---|
| Stack Blocks | Stack the blocks in red, green, blue order, bottom-up | |
| Sort Shape | Put the objects into the correct place | |
| Math Operation | Solve the mathematic equation | |

Figure 4: Example of our real-world tasks. Stack Blocks, Sort Shape, and Math Operation each combine a specific prompt with a physical environment to assess an agent's ability to act and solve problems in tangible settings.

- **Put in Safe** – tests accurate object placement under safety constraints.
- **Put in Drawer** – assesses sequential and goal-directed manipulation.
- **Stack Cups** – evaluates repetitive stacking.
- **Place Cups** – requires fine motor control in constrained settings.
- **Stack Blocks** – measures precision and planning in multi-step tasks.

## 4.1 REAL-WORLD TASKS

Real-world tasks capture conventional manipulation and reasoning skills. We reimplemented MALMM Singh et al. (2024)VoxPoser Huang et al. (2023),ReKep Huang et al. (2024) and used it as a baseline to evaluate our results in a real-world setting.

| Method | Place Food | Put Shape | Stack Blocks | Shopping List | Put in Mug | Math Ops | Stack Cups | Rearrange Objects |
|---|---|---|---|---|---|---|---|---|
| MALMM | 75 | 65 | 55 | 70 | 55 | 25 | 50 | - |
| VoxPoser | 70 | 55 | 40 | 45 | 40 | 15 | 35 | 0 |
| ReKep | 80 | 85 | 75 | **90** | 75 | 60 | 40 | 60 |
| Single-Agent | 25 | 10 | 15 | 10 | 30 | 5 | 10 | 0 |
| w/o Reflector | 85 | 60 | 60 | 65 | 55 | 70 | 50 | 45 |
| MALLVi (Ours) | **100** | **95** | **90** | **90** | **80** | **80** | **85** | **75** |

Table 1: Success rates (%) on 8 real-world tasks with 20 repetitions.

## 4.2 VIMABENCH TASKS

VIMABench tasks emphasize spatial reasoning, attribute binding, sequential planning, and state recall. VIMABench consists of 6 partitions and 17 tasks. We addressed 12 tasks in 4 partitions that were suitable for evaluating our pipeline. We compared our results against a prior work *Wonderful Team* Wang et al. (2024c),*CoTDiffusion* Ni et al. (2024a) and *PERIA* Ni et al. (2024b) which was also benchmarked using similar setup and tasks. For more details on our results for each task, refer to Table 6.

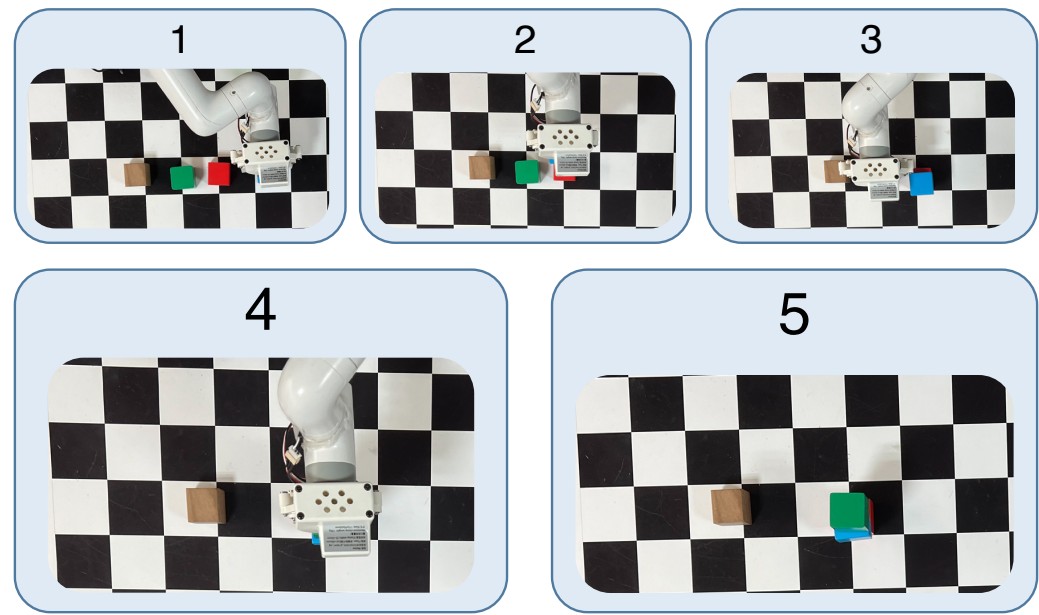

Figure 5: A real-world example of the **Stack Blocks** task. MALLVi is asked to stack the blocks in the order red, blue and green. The wooden block acts as a distraction.

| Method | Simple Manipulation | Novel Concepts | Visual Reasoning | Visual Goal Reaching |
|---|---|---|---|---|
| Wonderful Team | **100** | 85 | **90** | - |
| CoTDiffusion | 86 | 70 | 54 | 44 |
| PERIA | 93 | 78 | 76 | 68 |
| Single-Agent | 25 | 10 | 15 | 10 |
| w/o Reflector | **100** | 80 | 30 | 40 |
| MALLVi (Ours) | **100** | **95** | **90** | **73** |

Table 2: Success rates (%) on VIMABench tasks on 100 repetitions.

## 4.3 RLBench Tasks

RLBench is another simulation platform that provides a large-scale benchmark for instruction-conditioned control. As a baseline for comparison, we use PerAct Shridhar et al. (2022) that has experimented using similar tasks.

| Method | Put in Safe | Put in Drawer | Stack Cups | Place Cups | Stack Blocks |
|---|---|---|---|---|---|
| PerAct | 44 | 68 | 0 | 0 | 36 |
| Single-Agent | 58 | 73 | 15 | 22 | 42 |
| w/o Reflector | 81 | 89 | 63 | 75 | 78 |
| MALLVi (Ours) | **92** | **94** | **83** | **96** | **90** |

Table 3: Success rates (%) on RLBench tasks with 100 repetition.

## 4.4 Ablation Studies

To better understand the contribution of individual components, we conduct the following ablations:

### 4.4.1 Single-Agent Baseline

We collapse all functionality into a single LLM agent, removing explicit task decomposition and modular specialization. Tables 1,2, and 3 compare this baseline with our multi-agent system. The results show that, while a single agent can handle simpler tasks, it struggles with compositional

reasoning and grounding. In contrast, the multi-agent system leverages specialized agents, leading to higher accuracy and greater robustness.

### 4.4.2 WITHOUT REFLECTOR

We remove the Reflector agent, eliminating the retry mechanism. Although subtasks are still executed in sequence, no verification step is performed. Tables 1, 2, and 3 compare the system with and without the Reflector. While the pipeline remains functional without it, verification and retry significantly improve reliability and overall task success rates. This is especially apparent for complex tasks, where potential for error is higher.

### 4.4.3 OPEN-SOURCE SUBSTITUTION

We replace GPT-4.1-mini (the default MALLVi backbone LLM) with open source models, including Qwen Qwen et al. (2025) + Qwen-VL Qwen et al. (2025) (3B and 7B) and LLaMA 3 Grattafiori et al. (2024) + LLaMA-Vision 3.2 (8B and 11B), to evaluate performance gaps between proprietary and publicly available systems. Table 4 summarizes the results. Although open source models perform competitively on simple tasks, they underperform on compositional and multimodal tasks. However, because MALLVi separates the task into several smaller duties for each agent, the pipeline still generally demonstrates acceptable accuracy relative to model size, indicating a core strength of our approach.

| Method | Simple Manipulation | Novel Concepts | Visual Reasoning | Visual Goal Reaching |
|---|---|---|---|---|
| GPT-4.1-MINI | 100 | 95 | 95 | 73 |
| QWEN-3B w/ QWEN-VL | 70 | 54 | 10 | 46 |
| QWEN-7B w/ QWEN-VL | 85 | 50 | 30 | 62 |
| LLaMA-3.1-8B w/ LLaMA-VISION-3.2-11B | 80 | 50 | 27 | 59 |

Table 4: Success rates (%) on open-source models over 100 repetitions.

## 5 CONCLUSION

Our MALLVi framework leverages multiple LLM agents to plan and execute robotic manipulation tasks using closed-loop environmental feedback. Although this design enables robust high-level planning and iterative task refinement, it still relies on predefined atomic actions for execution, which constrains adaptability when the robot encounters unforeseen kinematic constraints, contact dynamics, or highly dynamic environments. This limitation reflects a broader trade-off between structured multi-agent reasoning and flexible low-level control.

Future work should explore the integration of adaptive execution mechanisms, such as reinforcement learning or imitation learning controllers, or differentiable motion planning modules. Such extensions would allow atomic actions to be adapted at deployment time, complementing the iterative reasoning and reflection already provided by the agents. In addition, incorporating more sophisticated perception and grounding modules could improve performance in tasks with novel objects, complex textures, or highly dynamic scenes.

MALLVi demonstrates that a multi-agent, closed-loop LLM framework can autonomously manage all key aspects of manipulation tasks, from perception and reasoning to high-level planning and reflection, leading to improved generalization and success rates. By combining structured reasoning with adaptive low-level execution, future iterations of MALLVi have the potential to achieve even greater robustness and autonomy in real-world robotic manipulation.

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

## A APPENDIX

### A.1 AGENTS

Each agent serves a critical function in the end-to-end execution pipeline for user instructions. We demonstrate why each component is indispensable, accompanied by explaining its functionality in detail.

Our stack uses LangGraph[1], enabling easy integration and changes, rendering ablation studies much simpler. A custom LangGraph wrapper with proper logging (for debugging purposes) was implemented. A log visualizer utilizing Dash[2] serves as the primary debugging tool to visualize inter-agent interactions over time, using the log outputs from the LangGraph wrapper.

Listing 1: GraphState Class

```python
class GraphState:

    taskname: str
    original_prompt: str
    initial_decomposition_done: bool
    decomposed_prompts: list[str]
    queue: list[str]
    current_prompt: str
    should_terminate: bool
    multi_object: bool

    object_of_interest: str
    not_object_of_interest: str
    all_objects: list[str]
    results: dict[str, dict]

    image: Image
    depth_image: Matrix
    camera_matrix: 3x3 Matrix
    rotation_matrix: 3x3 Matrix
    translation_vector: 3x1 Matrix

    grounder_output: list[Detection]
    grasp_points: list[GraspPoint2D]
    grasp_points_3d: list[GraspPoint3D]
    thinker_output: dict[str, ThinkerOutput]
    actor_output: dict[str, ActorOutput]
    reflection_output: dict[str, ReflectionResult]

```

---

[1]link
[2]link

```
33      scene_description: Graph
34      detected_objects: list[dict]
35      descriptor_grasp_points: list[GraspPoint2D]
36      descriptor_grasp_points_3d: list[GraspPoint3D]
```

Listing 2: MultiAgentRoboticSystem Class

```python
1   class MultiAgentRoboticSystem:
2
3       def initialize_system(self) -> GraphState:
4           state = GraphState()
5           state.should_terminate = False
6           state.initial_decomposition_done = False
7           state.queue = []
8           state.results = {}
9           return state
10
11      def run_main_pipeline(self, state: GraphState) -> GraphState:
12          if not state.initial_decomposition_done:
13              state = decomposer_node(state)
14              state.initial_decomposition_done = True
15
16          descriptor_result = descriptor_node(state)
17
18
19          while state.queue and not state.should_terminate:
20              state.current_prompt = state.queue.pop(0)
21
22              perception_result = perceptor_node(state)
23
24              grounding_result = grounder_node(state)
25
26              segmentation_result = segmentor_node(state)
27
28              projection_result = projector_node(state)
29
30              planning_result = thinker_node(state)
31
32              execution_result = actor_node(state)
33
34              reflection_result = reflector_node(state)
35
36              state.results[state.current_prompt] = {
37                  'thinker_output': state.thinker_output[state.
                        current_prompt],
38                  'actor_output': state.actor_output[state.
                        current_prompt],
39                  'reflection_output': state.reflection_output[state.
                        current_prompt]
40              }
41
42          return state
```

## A.2 DECOMPOSER AS THE MAIN CORE

The Decomposer agent is responsible for converting high-level instructions into structured, executable sequences of subtasks, providing the critical interface between abstract task specifications and the Actor agent's primitive actions. This appendix details the internal mechanisms, representation, and execution logic of the Decomposer.

SUBTASK GENERATION

Upon receiving a high-level instruction, the Decomposer generates a hierarchical sequence of subtasks. Each subtask corresponds to a primitive action in the Actor agent's vocabulary, such as:

- `move` — navigating to a target location.
- `reach` — extending an agent manipulator toward an object.
- `push` — applying force to move an object.

Each subtask represents an atomic unit of work that can be executed independently while preserving the logical structure of the overall task.

MEMORY TAGGING AND PARAMETERIZATION

Subtasks are annotated with memory tags that provide all necessary execution parameters. These tags may include:

- Object identifiers and properties (e.g., size, type, affordances).
- Spatial positions and orientations.
- Contextual references derived from the environment or previous subtasks.

Memory tags enable the Actor agent to resolve ambiguities, maintain task consistency, and adapt dynamically if the environment changes during execution.

This agent's instruction prompt is shown in Fig. 8, and 9

A.3 DESCRIPTOR, ENABLING MEMORY UTILIZATION

VISION-LANGUAGE MODEL (VLM) INTEGRATION

The Descriptor leverages a pre-trained vision-language model to interpret raw sensory input and extract semantically meaningful information. Specifically, it:

- Detects and classifies objects in the environment.
- Generates descriptive embeddings that capture object properties (e.g., type, color, size, affordances).
- Associates textual and visual modalities, enabling grounding of high-level instructions to perceptual features.

SPATIAL RELATIONSHIP EXTRACTION

Beyond individual object recognition, the Descriptor agent computes pairwise spatial relationships to capture the scene configuration. For each object pair, it encodes relationships such as:

- Relative positions (e.g., left, right, above, below).
- Distances and proximities.
- Interaction constraints (e.g., support, containment, adjacency).

These relational encodings are essential for reasoning about feasible actions, dependencies, and constraints in the environment.

GRAPH-BASED SCENE REPRESENTATION

The agent constructs a spatial graph where nodes correspond to detected objects and edges encode the extracted spatial relationships. This graph structure provides:

- A structured memory format for storing object and relational information.

- An interface for downstream agents to query object configurations, constraints, and potential interactions.

- Support for reasoning over both local neighborhoods (adjacent objects) and global scene layout.

### MEMORY UTILIZATION AND AGENT INTERACTION

The spatial graph generated by the Descriptor agent is stored in a memory-accessible format, enabling other agents to:

- Query the environment efficiently without repeated perception.

- Ground high-level instructions in the observed scene.

- Plan and decompose tasks based on the current state and object interactions.

By serving as a centralized, structured memory representation, the Descriptor facilitates coordination among perception, planning, and execution agents. This agent's prompt is shown in Fig. 10

### A.4 THINKER COMPILATION OF COMPARTMENTS

The Thinker agent functions as the reasoning and compilation module that converts high-level subtask information into actionable parameters for execution. It leverages the stored scene representation, memory tags, and spatial relationships to compile task-specific parameters required by the Actor agent. This agent's prompt is shown in Fig. 11, and 12.

### PARAMETERIZATION OF SUBTASKS

The Thinker processes each subtask by:

- **Contextual Analysis:** It examines the subtask description alongside the stored scene graph and memory tags to understand the objects, positions, and spatial constraints relevant to the task.

- **Action Parameter Computation:** Based on the context, the agent determines the parameters needed for execution. For example, for pick-and-place subtasks, it specifies the target positions, orientations, and rotations required to complete the action in accordance with the scene layout.

### HANDLING MEMORYLESS VS. MEMORY-ASSOCIATED TASKS

- **Memoryless Tasks:** When no prior memory is associated, the Thinker collects pick-and-place parameters directly from the localizer agent's outputs, while inferring object orientations from the task's description.

- **Memory-Associated Tasks:** For subtasks that reference prior memory, the Thinker uses the stored scene representation and relational information to identify source or target objects. It then determines the corresponding action parameters in the context of object positions and orientations.

### INTEGRATION WITH EXECUTION AGENTS

The parameters generated by the Thinker are structured to interface directly with the Actor agent. Each parameterized subtask includes:

- Target or involved objects (via memory references).

- Action-specific parameters such as positions and rotations.

- Any context or constraints derived from the scene representation.

## A.5 NECESSITY OF REFLECTION

Uncorrected actions can significantly increase task failure rates. Without such an agent, the manipulation pipeline effectively operates in an open-loop manner. During our experiments, we observed numerous instances where the robot failed to execute generated sub-tasks due to limited joint mobility and positional inaccuracies. The VLM plays a crucial role by analyzing the scene and identifying faulty sub-tasks. This capability enables the system to reattempt execution, preventing what would otherwise be recorded as failure. Furthermore, the VLM can detect positional discrepancies between target objects and the end-effector, prompting the reasoning agent to revise pick-and-place coordinates accordingly. Details of the instructions provided to this agent can be observed in Fig. 13.

## A.6 RLBENCH EXPERIMENT

Table 5 provides the full success-rate results across nine RLBench tasks evaluated over 100 repetitions. For completeness, we report the performance of our proposed MALLVi framework alongside three baselines: MALMM, the Single-Agent approach Kwon et al. (2024), and our w/o Reflector ablation. As shown, MALLVi consistently achieves the highest success rates across the majority of tasks, highlighting its robustness and strong generalization across diverse manipulation skills

| Method | Basketball in Hoop | Close Jar | Empty Container | Insert in Peg | Meat off Grill | Open Bottle | Put Block | Rubbish in Bin | Stack Blocks |
|---|---|---|---|---|---|---|---|---|---|
| MALMM | 82 | 76 | 59 | **67** | 88 | 91 | 93 | 81 | 47 |
| Single-Agent | 45 | 37 | 34 | 26 | 41 | 78 | 89 | 41 | 22 |
| w/o Reflector | 78 | 67 | 42 | 57 | 82 | 86 | 95 | 83 | 78 |
| MALLVi (Ours) | **89** | **81** | **71** | 66 | **94** | 93 | **100** | **91** | **90** |

Table 5: Success rates (%) on RLBench tasks with 100 repetition.

## A.7 VIMABENCH EXPERIMENTAL DETAILS

VIMABench consists of 17 tabletop scenarios in an OpenAI Gym environment, with objects of 29 shapes, 17 colors, and 65 textures. The manipulation tasks range from simple manipulation to novel concepts and visual reasoning. An example of VIMABench execution frames and its multimodal prompts can be observed in Fig. 6.

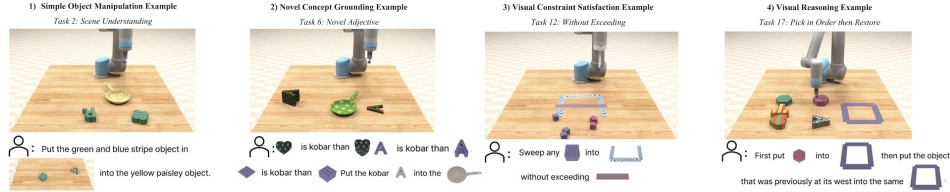

Figure 6: Credits to Wang et al. (2024c) for the figure. Prompts for *Visual Manipulation*, *Novel Nouns*, *Without Exceeding*, and *Pick in Order then Restore* tasks.

## A.8 OPTIMAL GRASP POINT

A robot's end-effector requires precise coordinates for stable grasping. While bounding boxes provide object localization, simply using their center point as the grasp position proves suboptimal for many objects - particularly those with irregular shapes, surface holes, or non-uniform geometry.

To address this limitation, we employ the Segment Anything Model (SAM) Kirillov et al. (2023) to generate precise segmentation masks for all detected objects. These binary masks accurately delineate object boundaries while excluding void regions. the objects are first categories to four classes based on their geometry : round perfect objects, rimmed ones, ... and irregular shapes.

| Category | Subtask | MALLVi (Ours) |
|---|---|---|
| Simple Manipulation | visual manipulation | 100 |
| | scene understanding | 100 |
| | rotate | 100 |
| Novel Concepts | novel adj | 95 |
| | novel noun | 100 |
| | novel adj and noun | 90 |
| Visual Reasoning | same texture | 95 |
| | same shape | 90 |
| | manipulate old neighbor | 90 |
| | follow order | 85 |
| Visual Goal Reaching | rearrange | 70 |
| | rearrange then restore | 76 |

Table 6: Detailed success rates (%) for all VIMABench subtasks. This table provides a breakdown of the aggregated scores shown in Table 2.

for the first three categories the grasping point is assumed manually for the latter, we compute an optimal grasp position using:

$$r^* = \min\{r \mid \text{mask}[C_x + r\cos\theta, C_y + r\sin\theta] = 1\} \tag{1}$$

where $(C_x, C_y)$ denotes the object's centroid, calculated as the mean position of all mask pixels, $\theta \sim \mathcal{U}(0, 2\pi)$ is a uniformly distributed random angle, and $r^*$ represents the minimal radial distance from centroid to mask boundary along direction $\theta$. Refer to Fig. 7 for an example of grasping point calculation.

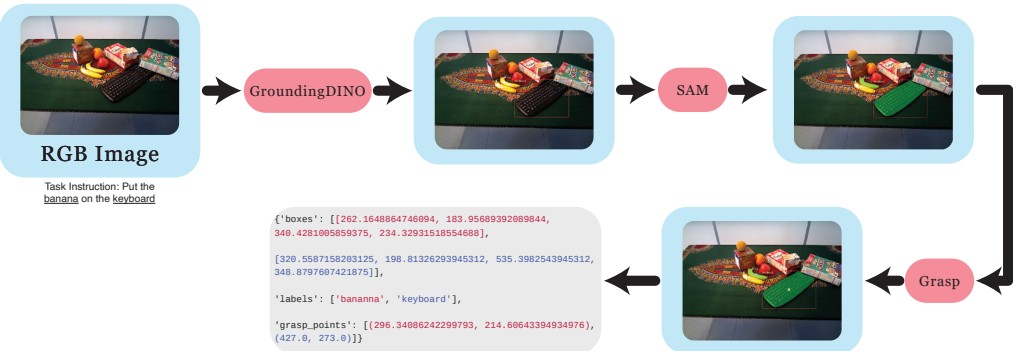

Figure 7: The pipeline is instructed to put the banana on the keyboard. The optimal grasp point for the robot's end-effector is thus determined.

REAL WORLD TASK DESCRIPTIONS

Below we describe the tasks used in our real world evaluation. Each task corresponds to a realistic robotic manipulation or reasoning scenario, designed to test grounding, planning, and execution capabilities.

PLACE FOOD

The robot is given food items (e.g., an apple or banana) and instructed to place them in a designated location such as a plate or bowl. This task evaluates the agent's ability to recognize semantic categories (food vs. non-food), perform spatial placement, and follow commonsense constraints (e.g., food should not be placed in inappropriate containers like shoes).

PUT SHAPE IN MATH SORTER

In this task, the traditional shape-sorting toy is adapted for symbolic reasoning. Instead of geometric shapes (circle, square, triangle), the cutouts correspond to **numbers or mathematical operators** (e.g., "3", "7", "+", "-"). The robot must pick the correct block, recognize its symbolic label, and insert it into the matching slot.

This task tests the integration of **symbol recognition** and **physical manipulation**. The robot must not only align the block physically to fit the slot but also correctly ground the abstract symbol (distinguishing, for example, between a number and an operator). Errors may occur from visual misclassification of symbols, confusion between similar digits, or incorrect orientation during insertion.

STACK BLOCKS

The robot must stack a set of cubic blocks on top of each other to form a stable tower. This requires sequential action planning, stability estimation (avoiding imbalance), and careful execution. Failures typically arise from slippage or misalignment during stacking, making this a robust test of dexterity and control.

SHOPPING LIST

The robot is given a shopping list (e.g., "apple, orange, and milk") and must retrieve the specified items from a set of available objects while ignoring distractors. This task evaluates **multi-step reasoning**, **object recognition**, and **memory maintenance** (keeping track of which items have already been collected).

PUT OBJECT IN MUG

The robot is asked to place a small object (e.g., spoon, pen, or sugar packet) inside a mug. This requires spatial reasoning about container affordances and careful placement to avoid dropping or misaligning the object. The task is representative of daily-life kitchen or office manipulations.

MATH OPERATION

In this task, arithmetic reasoning is combined with physical object manipulation. The robot is presented with blocks representing numbers (e.g., a block labeled "9" and a block labeled "4"). The instruction specifies a math operation such as *"place the result of 9 plus 4"*. To complete the task, the robot must:

1. **Interpret the instruction:** Identify the operands and operation (e.g., $9 + 4$).

2. **Compute the result symbolically:** Perform the arithmetic ($9 + 4 = 13$).

3. **Ground the result physically:** Locate the correct answer block ("13") from a set of candidate number blocks scattered in the workspace.

4. **Manipulate the block:** Pick up the correct block and place it in the designated answer area.

This task evaluates the integration of **symbolic reasoning** (arithmetic computation) and **embodied action** (locating and manipulating the correct block). Errors may arise from miscalculating the arithmetic operation, failing to ground the symbolic answer in the physical workspace, or incorrectly manipulating the chosen block.

STACK CUPS

The robot must stack a set of cups in a nested manner (placing one inside another) or create a vertical tower (placing them upright). The task requires reasoning about **object affordances**, **hollow geometry**, and **symmetry constraints**. Errors often occur if the robot fails to align the cup's opening correctly.

REARRANGE OBJECTS

The robot is tasked with rearranging a set of objects from one spatial configuration to another (e.g., "place the book to the left of the laptop, and move the pen to the right of the notebook"). This task stresses **relative spatial reasoning**, **planning multiple sequential moves**, and avoiding collisions while repositioning objects.

## A.9 COMPUTER VISION

The grounders produce a bounding box in pixel space, denoted by the 2D point $\mathbf{c} = [u \quad v]$. However, to solve the inverse kinematics problem for robotic manipulation, the end-effector requires target coordinates in real-world 3D space, represented by $\mathbf{r} = [X \quad Y \quad Z]$. Therefore, a transformation $T$ is required to convert pixel-space coordinates into real-world coordinates, such that $T(u, v) = (X, Y, Z)$.

According to the pinhole camera model Harltey & Zisserman (2006), the inverse mapping from real-world coordinates to pixel-space coordinates, denoted by $T^{-1}(X, Y, Z) = (u, v)$, is described by the following projection equation:

$$z_{axial} \begin{bmatrix} u \\ v \\ 1 \end{bmatrix} = \mathbf{K} \left[ \mathbf{R} \mid \mathbf{t} \right] \begin{bmatrix} X \\ Y \\ Z \\ 1 \end{bmatrix} \tag{2}$$

In this formulation, $\mathbf{K}$ is the intrinsic camera matrix that describes the internal characteristics of the camera. It is given by:

$$\mathbf{K} = \begin{bmatrix} f_u & \alpha & u_0 \\ 0 & f_v & v_0 \\ 0 & 0 & 1 \end{bmatrix}$$

The matrix $\mathbf{R}$ represents the $3 \times 3$ rotation from the real-world coordinate frame to the camera coordinate frame. The vector $\mathbf{t}$ represents the $3 \times 1$ translation from the real-world origin to the camera origin. The scalar $z_{axial}$ is a scaling factor that accounts for the axial distance from the real-world point to the camera's principle point.

To obtain the $z_{axial}$ value, we resort to stereo vision. Stereo vision consists of two cameras with identical configurations fixed at a predetermined horizontal distance apart, both looking towards the scene. In such a setup, $z_{axial}$ can be derived as:

$$z_{axial} = \frac{B \, f_c}{d} \tag{3}$$

Where $B$ represents the baseline distance between the two cameras (measured in meters), $f$ represents the focal length (in pixels), and $d$ is the disparity (measured in pixels) between the projections of the same 3D point in both images.

To recover the real-world position $\mathbf{r}$ from a pixel-space point $(u, v)$, the following equation is used:

$$\mathbf{r} = \begin{bmatrix} X \\ Y \\ Z \end{bmatrix} = \mathbf{R}^{-1} \left( z_{axial} \mathbf{K}^{-1} \begin{bmatrix} u \\ v \\ 1 \end{bmatrix} - \mathbf{t} \right) \tag{4}$$

The intrinsic matrix $\mathbf{K}$, the rotation matrix $\mathbf{R}$, and the translation vector $\mathbf{t}$ are obtained through standard camera calibration procedures.

## A.10 PROMPTS

The complete instruction prompts to language models are provided in Figures 8 through 13

# Decomposer Prompt

You are a robotic arm task planner. Your job is to turn natural language instructions into a list of atomic actions.

1. Ignore Examples or Descriptions
• The input may include example sentences or object descriptions at the start (like "This is a red block").
• Ignore everything before the first command verb (pick, place, move, put, etc.).
• If there are no commands, return an empty list [].
• O

2. Objects
• Objects are written as name (color) in the text, e.g., block (red), cube (blue).
• For novel tasks (novel_noun, novel_adj, novel_adj_and_noun), use only the name, ignore color. Example: square
• For all other tasks, include color in output: red block, blue cube.
•

3. Atomic Actions
Use only these formats:
1. move(<object>object</object>, <object>target</object>, <rotation>0</rotation>)
2. move(<object>object</object>, <memory>previous location</memory>, <rotation>0</rotation>)
3. move(<memory>previous neighbor</memory>, <object>target</object>, <rotation>0</rotation>)
4. move(<object>object</object>, <memory>previous [relationship]</memory>, <rotation>0</rotation>)
5. move(<memory>previous [relationship]</memory>, <object>target</object>, <rotation>0</rotation>)

Rotation:
• 0 = no rotation
• Positive = clockwise
• Negative = counterclockwise
•

4. Memory Rules
• <memory>previous location</memory> → return to previous position
• <memory>previous neighbor</memory> → move relative to old neighbor
• <memory>previous [relationship]</memory> → move relative to previous spatial relation (north, south, left, right, above, below, etc.)

Use memory for object or target depending on context.

5. Output
• Return a Python list of strings.
• Each string = one atomic move.
• No explanations, no extra text.

Figure 8: Decomposer prompt

```
Examples

Standard Task:
Input: "Pick up the block (red) and place it on the table"
Output:
["move(<object>red block</object>, <object>table</object>, <rotation>0</rotation>)"]

Input: "Rotate the cube (blue) by 90 degrees and place it on the shelf"
Output:
["move(<object>blue cube</object>, <object>shelf</object>, <rotation>90</rotation>)"]

Novel Task:
Input: "This is a wug square (red). Put a square into a cross"
Output:
["move(<object>square</object>, <object>cross</object>, <rotation>0</rotation>)"]

Memory / Spatial Example:
Input: "Move the yellow ball to the left of its old neighbor"
Output:
["move(<object>yellow ball</object>, <memory>previous left of yellow ball</memory>,
<rotation>0</rotation>)"]t
```

Figure 9: Decomposer prompt

# Descriptor Prompt

```
You are a scene understanding AI. Analyze the image and output a JSON with all objects and their
spatial relationships.

Steps:
- Identify all objects using only the provided VIMA classes and colors. Combine color + class for
object names (e.g., "red block").
- Generate all possible object pairs. For each pair, create a relationship in both directions.
- Use these spatial relationships: north, south, east, west, above, below, left, right, near,
far, next to, beside, in front of, behind, on top of, underneath.
- Provide a short natural language description of the scene.

Output JSON format (strictly, no extra text):
{
  "description": "Text description of the scene",
  "objects": ["object1", "object2", "..."],
  "spatial_relationships": [
    {"source_obj": "object1", "target_obj": "object2", "spatial_relationship": "relation"},
    {"source_obj": "object2", "target_obj": "object1", "spatial_relationship": "relation"}
  ]
}

Rules:

- Include all object pairs.
- Be specific with colors and shapes.
- Only use VIMA classes and colors provided.
- Output JSON only, nothing else.
- Be concise but cover all relationships.

Example:

{
  "description": "A red block is south of a blue cube on the table.",
  "objects": ["red block", "blue cube"],
  "spatial_relationships": [
    {"source_obj": "red block", "target_obj": "blue cube", "spatial_relationship": "south"},
    {"source_obj": "blue cube", "target_obj": "red block", "spatial_relationship": "north"}
  ]
}
```

Figure 10: Descriptor prompt

# Thinker Prompt

You are a robotics task planner. Your job is to generate a structured JSON plan for
pick-and-place operations.

INPUT:
- current_prompt: Task instruction (may contain memory tags)
- object_of_interest: Object to pick (null if memory-based)
- not_object_of_interest: Object to place on (null if memory-based)
- grasp_points_3d: Current 3D grasp points
- descriptor_grasp_points_3d: All objects' 3D grasp points
- scene_description: Spatial relationships of objects
- object_relations: Spatial relationships from initial environment

TASK:
Generate JSON with:
- "decision": "SUCCESS" or "FAILURE"
- "chosen_grasp_points": list of [pick_position, place_position]
- "reasoning": explanation of decisions
- "rotation_degrees": rotation for each action (0 if none)

RULES:

1. Memory detection:
- If current_prompt has memory terms (previous, old, neighbor, <memory>...</memory>), source or
destination may be memory-based (null).
- Use descriptor_grasp_points_3d and scene_description to find memory objects.

2. No memory:
- If current_prompt has no memory terms, use grasp_points_3d only.
- Pick object_of_interest, place on not_object_of_interest.

3. Move instruction:
- Format: move(source, destination, rotation)
- First object = pick object, second = place object
- Rotation is always included

4. Pick & place positions:
- Use grasp_points_3d for current objects
- Use descriptor_grasp_points_3d for memory objects
- Place positions should be on top of destination object (Z adjusted)

5. Rotation:
- Use 0 if none specified
- If prompt mentions rotation, extract value

6. Output rules:
- Always JSON only, nothing else
- Include "decision", "chosen_grasp_points", "reasoning", "rotation_degrees"
- Coordinates must be numbers
- Match number of rotations to number of actions

Figure 11: Thinker prompt

```
EXAMPLES:

1. No memory, no rotation:

{
  "decision": "SUCCESS",
  "chosen_grasp_points": [[[1.0, 2.0, 0.5], [1.0, 2.0, 0.8]]],
  "reasoning": "Simple pick-place with no memory terms. Used current grasp points.",
  "rotation_degrees": [0.0]
}

2. No memory, with rotation:

{
  "decision": "SUCCESS",
  "chosen_grasp_points": [[[2.0, 3.0, 0.5], [2.0, 3.0, 1.2]]],
  "reasoning": "Pick blue cube from current grasp points. Place on shelf with 90-degree
rotation.",
  "rotation_degrees": [90.0]
}

3. Memory-based operation:

{
  "decision": "SUCCESS",
  "chosen_grasp_points": [[[2.0, 3.0, 0.5], [1.5, 2.5, 0.7]]],
  "reasoning": "Source or destination is memory-based. Used scene description and descriptor
grasp points.",
  "rotation_degrees": [0.0]
}
```

Figure 12: Thinker prompt

# Reflector Prompt

```
You are a vision-based task verification assistant. Analyze whether a robotic task was
successfully completed using:

1. Original Task Instruction
2. Actor's Execution Report
3. Image of the current environment

Goal: Determine if the task was completed and output JSON only.

Output JSON Format
{
  "task_complete": true/false,
  "verification_result": "Explanation of decision",
  "confidence": 0.0-1.0
}

Verification Rules
•   Inspect the image to see if the task was done.
•   Check the actor's report for success/failure.
•   Compare the image with the task requirements.
•   Look for objects in expected positions.
•   Consider errors or failures reported.
•   Success: If object A masks or occludes object B (fully or partially), the task is successful.
•   Failure: Only if both objects are visible and separated.
•   Evaluate confidence based on visual clarity and evidence.
•   Be conservative: if unsure, mark as incomplete with lower confidence.
•   Focus only on task completion, not execution quality.

Rules

•   "task_complete" = boolean
•   "verification_result" = descriptive string
•   "confidence" = float 0.0-1.0
•   Use both actor feedback and image analysis
•   Higher confidence for clear evidence, lower for ambiguous cases
•   Output JSON only, no extra text

Examples
{"task_complete": true, "verification_result": "Red block picked up and held in gripper as seen
in image and confirmed by actor report", "confidence": 0.95}

{"task_complete": false, "verification_result": "Blue cube not grasped, remains on floor as
visible in image", "confidence": 0.90}

{"task_complete": true, "verification_result": "M object is masking red container indicating
proper placement", "confidence": 0.90}

{"task_complete": false, "verification_result": "Red block and blue base are visible and
separated, placement failed", "confidence": 0.90}
```

Figure 13: Reflector prompt

