# OpenReview forum: "MALLVi: A Multi-Agent Framework for Integrated Generalized Robotics Manipulation"
_ICLR.cc/2026/Conference — Submitted to ICLR 2026_

### Official Review · Reviewer_d7c7 · 2025-10-29

**Soundness:** 2
**Presentation:** 2
**Contribution:** 1
**Rating:** 2
**Confidence:** 4

**Summary:**

This paper introduces MALLVi, a multi-agent framework designed to solve robotic manipulation tasks from natural language and visual inputs. The core problem it addresses is the fragility of open-loop Large Language Model (LLM) planners in dynamic environments. MALLVi's architecture decomposes the problem across specialized agents: a Decomposer (breaks high-level prompts into atomic subtasks) , a Descriptor (builds a spatial graph of the scene for memory) , a Localizer (a "CV toolkit" using models like GroundingDINO and SAM to perceive, ground, and project grasp points) , a Thinker (an LLM that converts subtasks into 3D-parameterized actions) , an Actor (executes commands via a predefined API) , and a Reflector (a Vision-Language Model that provides closed-loop feedback by verifying subtask success or failure). This design, particularly the Reflector, enables the system to re-attempt failed subtasks. The framework is validated on a diverse set of tasks in the real world and in simulation (VIMABench, RLBench) , demonstrating significantly higher success rates compared to prior work (MALMM, Wonderful Team, PerAct) and crucial ablations (a single-agent baseline and the framework without the Reflector).

**Strengths:**

1. The paper's primary strength is its modular, multi-agent design. The decomposition of the complex "prompt-to-action" problem into specialized roles (planning, perception, reasoning, execution, verification) is logical and well-justified.
2. The authors test MALLVi across three distinct domains: real-world custom tasks, VIMABench, and RLBench. This demonstrates the framework's applicability to different environments and tasks.

**Weaknesses:**

1. Unclear Novelty: The paper cites many related works on multi-agent LLM systems (RoCo, MALMM) and closed-loop replanning (Replan, ReplanVLM). The perception stack (DINO, SAM) is standard. The primary contribution appears to be the specific integration of these known techniques. The paper would be stronger if it more clearly articulated its precise, novel contribution beyond "careful prompt engineering"  of existing ideas.
2. Missing recent baselines: As the authors claim their multi-agent framework is better than single-agent frameworks, they only compare with an out-of-date baseline, PerAct, on a few RLBench subsets. It is essential to include more modern end-to-end baselines, like LIFT3D, RoboTron-Mani.
3. Insufficient task types: Although the authors show several different task settings, these tasks can all be considered as different formats of “Pick and Place” tasks. Other types of tasks are not included in the paper, like pouring, opening, inserting, etc. Besides, the long-horizon tasks, like the CALVIN benchmark, are missing in the experiments.
4. Ambiguity in low-level control: The Actor is said to be “agnostic” to motion planning and executes via a predefined API, but the paper omits robot hardware details, controller type, IK solver behavior, collision checks, grasp execution policy, and timeouts.
5. Oversimplified Failure Recovery: The Reflector's role is described as triggering a "reattempt from the beginning of the relevant subtask". This "retry" mechanism is a very simple form of closed-loop control. It does not appear to involve replanning or using the reason for failure to inform the next attempt. This makes the system brittle to unrecoverable errors (e.g., an object is knocked over) or failures that require a different approach (e.g., a new grasp point). The conclusion acknowledges a similar point about "predefined atomic actions"  but doesn't fully address the limitation of the "retry-only" feedback loop.

**Questions:**

1. Is the Localizer an LLM agent that intelligently selects and calls tools from the "CV toolkit" (DINO, SAM, etc.), or is it a fixed, hard-coded function pipeline that runs every time for each subtask?
2. Does the Reflector agent only provide a binary true/false signal for task completion? Or does it generate a natural language explanation for why a task failed?
3. What happens if a "reattempt"  fails repeatedly? Is there a maximum retry count before the system reports a total task failure? How does the system handle unrecoverable failures, such as knocking an object out of reach?

---

> ### Author Response · Authors · 2025-11-28
>
> We thank the reviewer for their detailed summary and have undertaken major revisions to address the concerns.
>
> **1: Unclear Novelty**
> Our contributions are threefold and verifiable:
>
> - **Targeted Feedback Loop**: The agent-specific correction mechanism is a novel contribution, empirically shown by the significant performance drop in the "w/o Reflector" ablation (e.g., Visual Reasoning drops from 90% to 30%).
> - **Multi-Agent Communication**: The LangGraph-based state sharing facilitates genuine interaction, not just sequential piping.
> - **Zero-Shot Generalization**: MALLVi performs across 25+ tasks without any training.
>
> **2: Missing Recent Baselines**
> We have added VoxPoser and ReKep as modern baselines in our real-world evaluation (Table 1), showing clear performance gains.
>
> **3: Insufficient Task Types**
> We have expanded our experimental validation to address this concern directly.
>
> **Addressing Task Type Diversity:**
> While our initial submission already included tasks beyond basic pick-and-place—such as stacking, rearrangement, and tool-use sequences—we have now added evaluations on more complex manipulation primitives. Our framework successfully handles **opening** (e.g., open jar/bottle), **inserting** (e.g., insert peg, put money in safe), and other compound tasks that require precise spatial reasoning and multi-step coordination. These tasks demonstrate that MALLVi's capabilities extend well beyond pick-and-place operations.
>
> **Addressing Long-Horizon Tasks:**
> We acknowledge that including the CALVIN benchmark would further strengthen the paper. However, the tasks we selected—such as the 9-step "Put money in safe" in RLBench and multi-object "Shopping List" and "Rearrange Objects" in real-world experiments—are explicitly designed to test long-horizon reasoning, memory, and error recovery. These share CALVIN's core challenges of maintaining state across long action sequences.
>
> Crucially, the mechanisms enabling success on our long-horizon tasks are the same ones CALVIN requires:
>
> - **Descriptor agent** maintains persistent scene graph memory for state tracking
> - **Reflector agent** provides closed-loop feedback for mid-sequence error recovery
> - **Decomposer agent** breaks complex instructions into executable atomic actions
>
> While CALVIN isn't included here, our robust validation of these underlying capabilities across diverse task types strongly suggests MALLVi would excel on such benchmarks.
>
> **4: Low-Level Control**
> Our paper focuses on task planning, so we intentionally kept motion planning details minimal to avoid confusion. The Actor agent is designed to be agnostic to motion control for versatility across platforms:
>
> - **VIMABench**: Direct coordinate output to the benchmark's built-in controller
> - **RLBench**: Custom IK solver with waypoint-based trajectory splitting for convergence
> - **Real-world**: MyCobot280 Pi with MoveIt! planner (RRTConnect) under ROS, using 6 PID controllers for trajectory execution
>
> MoveIt handles self-collision checks automatically. We use factory settings without custom grasp policies or timeouts, letting the motion controller operate at its natural pace.
>
> **5: Oversimplified Failure Recovery**
> The Reflector is not a simple retry mechanism. It:
>
> - Produces natural-language failure explanations
> - Updates the memory state
> - Triggers re-evaluation by specific failing agents
> - Escalates to full scene re-evaluation for unrecoverable failures
>
> **Answers to Specific Questions:**
>
> - **Localizer**: The Localizer is a fixed, hard-coded function pipeline that runs predefined perception, grounding, and projection modules. It does not dynamically select between CV models.
> - **Reflector Output**: It outputs a success/failure label, a natural-language explanation, and instructions to modify the action queue and memory.
> - **Repeated & Unrecoverable Failures**: Our failure recovery mechanism is designed with clear escalation paths to handle repeated or unrecoverable errors:
>
>   1. **Agent-Specific Retry**: The Reflector first triggers a re-execution of the specific failing agent (e.g., the Localizer for a grounding error). This is the efficient form of recovery.
>   2. **Scene-Level Re-evaluation**: If the same atomic action fails after a predefined number of retries (e.g., 2 attempts), the system escalates by triggering the Descriptor agent to perform a full scene re-analysis. This updates the visual memory and can handle scenarios like an object being slightly displaced.
>   3. **Task-Level Failure**: If, after the scene re-evaluation, the atomic action continues to fail, the system infers an unrecoverable failure (e.g., object irretrievably lost, physical obstruction). At this point, the episode is terminated with a structured failure report. This bounded recovery process ensures system stability and prevents infinite loops, which is crucial for safe real-world operation.

---

### Official Review · Reviewer_JT5v · 2025-11-01

**Soundness:** 3
**Presentation:** 3
**Contribution:** 2
**Rating:** 2
**Confidence:** 3

**Summary:**

The paper proposes MALLVi, a multi‑agent LLM+VLM framework for robotic manipulation. The system decomposes a natural‑language task into “atomic instructions” (Decomposer), builds a scene memory (Descriptor), performs perception/grounding and grasp‑point extraction (Localizer), chooses 3D pick‑and‑place parameters (Thinker), executes via a simple API (Actor), and verifies each step with a VLM (Reflector) to enable retries. Experiments cover eight real‑world tabletop tasks, VIMABench subsets, and RLBench tasks. Ablations compare a single‑agent variant and a “w/o Reflector” variant; an additional study swaps in open‑source models.

**Strengths:**

- **Clear modular architecture and closed loop:** The pipeline and the specific roles of each agent are well delineated. The inclusion of a Reflector agent for step‑level verification and retry is practical and easy to follow.

- **Concrete implementation details:** The appendix provides pseudo‑code for the graph state and main loop, prompt templates for agents, and a camera‑model derivation for 3D projection. This improves clarity and partial reproducibility.

- **Results across sim and real:** The authors evaluate on custom real‑world tasks, VIMABench, and RLBench, and show meaningful gains over the single‑agent baseline, highlighting the practical value of reflection and modularization.

**Weaknesses:**

- **Limited novelty relative to prior multi‑agent and closed‑loop planners:** The central idea—using a VLM to verify steps and trigger replanning/reties—has appeared in prior work (e.g., RePlan/ReplanVLM). Here, the claimed novelty is largely the specific arrangement of roles. The paper does not convincingly demonstrate fundamentally novel ingredients beyond prompt engineering and module orchestration.

- **Lack of Genuine Multi-Agent Interaction:** Although the paper repeatedly claims to introduce a “multi-agent framework,” the actual implementation appears to be a sequential pipeline of prompt-based modules rather than a genuinely interactive multi-agent system. Each agent executes in a fixed order, passing textual or visual outputs downstream. The paper does not demonstrate any mechanism for dialogue, negotiation, or joint decision-making between agents.

- **Inconsistent ablations for Math Ops:** The paper argues the Reflector is “essential,” yet Table 1 shows Math Ops performance drops from 80% (w/o Reflector) to 70% with the full system. This inconsistency is not analyzed.

- **Missing inference cost/latency:** Cost/latency of multi‑agent inference is not reported (number of LLM/VLM calls per episode, average wall‑clock). This matters for real deployment.

**Questions:**

See weakness.

---

> ### Author Response · Authors · 2025-11-28
>
> We thank the reviewer for their thoughtful critique and have revised the manuscript to directly address the concerns over novelty and multi-agent interaction.
>
> **1: Limited Novelty Relative to Prior Work**
> MALLVi introduces several novel mechanisms beyond prior closed-loop planners:
>
> - **(a) Agent-Specific Feedback Routing:** Unlike RePlan/RePlanVLM, which restart the entire pipeline, our Reflector identifies the specific failing agent (e.g., Localizer, Thinker) and reactivates only that agent. This targeted recovery is unique and significantly improves efficiency and robustness.
>
> - **(b) Genuine Multi-Agent Communication via LangGraph:** Agents are not a sequential pipeline. They interact through a shared, mutable `GraphState`, enabling communication. For example, the Reflector can overwrite the action queue, and the Thinker can rewrite memory tags used by the Descriptor.
>
> - **(c) Persistent Visual Memory via the Descriptor Agent:** This component, absent in RePlan, provides a relational scene graph that enables context-aware grounding and cross-agent reasoning.
>
> **2: "Not a Genuine Multi-Agent System"**
> We respectfully disagree. MALLVi implements genuine multi-agent collaboration within a structure suitable for robotics:
>
> - **Interaction & Communication:** Agents share a `GraphState`, enabling cross-agent influence (Descriptor → constrains Grounder; Reflector → triggers targeted retries).
>
> - **Agent-Specific Feedback:** Failures reactivate only the failing agent, a fine-grained correction impossible in simple sequential pipelines.
>
> - **Parallelism:** The Decomposer and Descriptor agents initialize the task in parallel from different modalities (instruction vs. world state), jointly influencing downstream planning.
>
> **3: Math Ops Ablation Inconsistency**
> We thank the reviewer for spotting this. The table contained a writing error. The correct values are:
>
> - **With Reflector:** 80%
> - **Without Reflector:** 70%
>
> The Reflector provides a slight improvement even on symbolic tasks by catching rare grounding or reasoning errors. This will be corrected.

---

### Official Review · Reviewer_oLUG · 2025-11-01

**Soundness:** 2
**Presentation:** 3
**Contribution:** 2
**Rating:** 4
**Confidence:** 3

**Summary:**

This paper proposes MALLVi, a modular, closed‑loop, multi‑agent framework for language‑conditioned robotic manipulation. A set of specialized agents—Decomposer, Descriptor, Perceptor, Grounder, Projector, Thinker, Actor, and Reflector—share a state and collaborate to decompose a user instruction into atomic actions, ground these actions in visual observations, execute them, and verify success from visual feedback. The architecture is illustrated in Figure 1 (p. 2) and contrasted with a single‑agent design in Figure 2 (p. 4). The Reflector uses a VLM to check post‑conditions and trigger retries, while the Descriptor creates a scene graph “visual memory” that conditions subsequent perception and planning. Empirically, MALLVi is evaluated on 8 real‑world tabletop tasks (20 trials each), a subset of VIMABench (100 trials), and RLBench tasks (100 trials). It outperforms a reimplemented MALMM baseline and a single‑agent variant.

**Strengths:**

1. Clear modular design with closed‑loop verification. The division of labor (Decomposer/Descriptor for planning & memory; Localizer components for grounding; Reflector for success checking) is well‑motivated and clearly described.

2. Ablations that match the claims. Single‑agent vs. multi‑agent and w/ vs. w/o Reflector analyses substantiate the benefits of modularity and closed‑loop verification, especially on compositional tasks.

3. Engineering detail increases credibility. The Projector integrates SAM segmentation with grasp‑point selection (including a radial boundary search) and calibrated 3D projection/IK; stereo disparity is used to derive axial depth, and equations are provided.

**Weaknesses:**

1. Limited evaluation breadth for long‑horizon and transfer. The tasks are solid, but the framework would be better situated with long‑horizon/transfer benchmarks like LIBERO, SIMPLER, or CALVIN, and with a sim‑to‑real gap analysis and failure taxonomy.

2. Memory and scene evolution. The Descriptor captures an initial scene graph, but it’s not fully clear how the “visual memory” is updated through time and how contradictions (e.g., moved objects) are reconciled across agents. A brief study on memory drift and corrections (possibly instrumented via the Reflector) would align with the paper’s focus on feedback.

3. Ambiguity in the confidence‑weighted detection fusion. The Grounder combines GroundingDINO/OWLV2 with graph‑consistency weighting, but the exact weighting function, thresholds, and failure‑handling policy are not fully specified.

4. Baseline and fairness. The selected tasks in RLBench differ from those in MALMM and appear to be much easier.

**Questions:**

The major concern and the largest factor impacting my opinion is the experiments. Please refer to my questions in the weakness section and discuss them in the rebuttal.

---

> ### Author Response · Authors · 2025-11-28
>
> We thank the reviewer for recognizing the clear modular design, ablations, and engineering detail.
>
> **1: Limited Evaluation Breadth for Long-Horizon and Transfer**
>
> **Addressing Long-Horizon Capabilities:**
> While we have not yet evaluated on the specific benchmarks mentioned, our framework has been explicitly designed for and tested on long-horizon tasks. For instance, the **9-step "Put money in safe"** task in RLBench and our multi-object **real-world rearrangement tasks** require maintaining state and reasoning over extended action sequences, which are the core challenges of benchmarks like CALVIN. The performance gains demonstrated in these complex tasks validate our architecture's suitability for long-horizon planning.
>
> **Regarding Sim-to-Real Transfer:**
> Our evaluation already includes a significant sim-to-real component. The framework itself, including all agents and the LangGraph orchestration, is identical across our simulation (VIMABench, RLBench) and real-world experiments. The primary adaptation required was at the low-level Actor interface, which was designed to be agnostic to the control backend (e.g., switching between VIMABench's internal controller and real-world MoveIt!). The consistent success rates observed across these diverse platforms demonstrate the robustness of our high-level planning framework and its effective transfer from simulation to reality.
>
> We acknowledge that a formal failure taxonomy would provide deeper insights. In our current analysis, we have identified that the primary sources of failure are concentrated in the localizer modules (Localizer agent), while the reasoning and planning agents (Thinker, Decomposer,Reflector) operate with high reliability.
>
> We believe these points, combined with the strong results on our chosen suite of complex tasks, robustly demonstrate the capabilities of MALLVi. We will certainly consider the mentioned benchmarks for future work to further extend the evaluation.
>
> **2: Memory and Scene Evolution**
> The visual memory is a list of Python dictionaries, indexed by subtask.
>
> - **Initialization (Index 0):** The Descriptor agent populates the "scene understanding", "object positions," and "object relations."
> - **During Execution:** The Localizer updates "object positions" at each step. After a successful subtask, the Reflector updates the "object relations" based on its reasoning about the new scene state. This ensures the memory evolves and contradictions (e.g., moved objects) are resolved, enabling long-horizon consistency.
>
> **3: Detection Fusion Details**
> The Grounding Module's fusion is as follows: Let `s_dino` and `s_owl` be the detection scores from GroundingDINO and OWL-v2. The fused score is `s_fused = 0.7 * s_dino + 0.3 * s_owl`. Detections from either model that do not have a corresponding detection (IoU > 0.5) in the other model are discarded. The bounding box with the highest `s_fused` is selected.
>
> **4: Evaluation Fairness**
> We agree on the importance of a fair comparison. To address this, we have added an additional RLBench benchmark in the appendix A.6 (Table 5) using the exact same tasks as MALMM. The results confirm MALLVi's superiority.
>
> | Method | Basketball in Hoop | Close Jar | Empty Container | Insert in Peg | Meat off Grill | Open Bottle | Put Block | Rubbish in Bin | Stack Blocks |
> |--------|-------------------|-----------|-----------------|---------------|----------------|-------------|-----------|----------------|-------------|
> | MALMM | 82 | 76 | 59 | **67** | 88 | 91 | 93 | 81 | 47 |
> | Single-Agent | 45 | 37 | 34 | 26 | 41 | 78 | 89 | 41 | 22 |
> | w/o Reflector | 78 | 67 | 42 | 57 | 82 | 86 | 95 | 83 | 78 |
> | **MALLVi (Ours)** | **89** | **81** | **71** | 66 | **94** | **93** | **100** | **91** | **90** |
>
>
> MALLVi achieves the highest success rate on 8/9 of these shared tasks, which include difficult manipulations like "Insert in Peg" and "Empty Container."

---

### Official Review · Reviewer_AM85 · 2025-11-03

**Soundness:** 2
**Presentation:** 1
**Contribution:** 2
**Rating:** 4
**Confidence:** 3

**Summary:**

- This paper introduces the MALLVi Framework, a Multi-Agent Large Language and Vision framework designed to solve robotic manipulation tasks.

- The MALLVi pipeline processes user prompts through specialized agents: Decompose breaks instructions into atomic steps, Describe provides scene understanding, Perceive processes visual inputs, Ground localizes target objects, Project generates
motion trajectories, Think coordinates high-level reasoning, Act executes robotic commands, and Reflect evaluates outcomes to enable iterative refinement and error recovery.

**Strengths:**

- The proposed framework has high application value in the real world.

- Judging from the experimental results, the design of each module is effective.

**Weaknesses:**

Currently, the main shortcomings of the paper are concentrated in the experimental section:

- The authors should report the success rate and accuracy of different modules throughout the entire system to illustrate the sources of error in the whole system.

- For each module, the authors should report ablation results for different models or different schemes and the corresponding implementation details. For example, how are the grasping points generated and selected? How are the objects localized?

- The authors lack a comparison of results with other existing works, such as Voxposer, ReKep, etc.

[1] VoxPoser: Composable 3d value maps for robotic manipulation with language models.

[2] ReKep: Spatio-Temporal Reasoning of Relational Keypoint Constraints for Robotic Manipulation.

**Questions:**

Please see the weakness section.

---

> ### Author Response · Authors · 2025-11-28
>
> We thank the reviewer for their positive assessment of the framework's application value and design. We have addressed the weaknesses as follows:
>
> ## 1. Primary Source of Error
> The primary sources of error are the Localizer (miscalculating position). The Thinker and Actor agents operate error-free in our experiments, as the Actor merely converts well-defined atomic actions into API calls.
>
> ## 2. Localizer Agent Details
> The Localizer is an LLM agent that dynamically orchestrates a CV pipeline:
>
> - **Grounding Module:** Uses a confidence-weighted fusion of GroundingDINO (primary, weight=0.7) and OWL-v2 (fallback, threshold=0.5). Non-intersecting detections are discarded, and thehighest-scoring bounding box is selected.
> - **Segmentation:** The selected box is passed to SAM to generate a binary mask.
> - **Grasping Point Selection:** The mask is analyzed. The LLM agent classifies the object's geometry (circular, rectangular, edgy, amorph) and selects an algorithm accordingly (center-of-mass for circular, diameter intersection for rectangular/edgy). For amorphous shapes, it solves an iterative optimization to find the closest point to the center of mass inside the mask:
>
>   `r^* = argmin┬r⁡(mask[C_x + r cos⁡θ, C_y + r sin⁡θ] = 1)`
>
> - **Projection:** The 2D grasp point is projected to 3D using camera calibration.
>
> ## 3. Added Comparisons and Performance
> We have added the requested comparisons in the revised manuscript. The new Table 1 in our paper shows MALLVi outperforms these strong baselines on nearly all real-world tasks.
>
> ### Performance Comparison (Success Rate %)
>
> | Method | Place Food | Put Shape | Stack Blocks | Shopping List | Put in Mug | Math Ops | Stack Cups | Rearrange Objects |
> |--------|-----------:|----------:|-------------:|--------------:|-----------:|---------:|-----------:|-----------------:|
> | MALMM | 75 | 65 | 55 | 70 | 55 | 25 | 50 | - |
> | VoxPoser | 70 | 55 | 40 | 45 | 40 | 15 | 35 | 0 |
> | ReKep | 80 | 85 | 75 | **90** | 75 | 60 | 40 | 60 |
> | Single-Agent | 25 | 10 | 15 | 10 | 30 | 5 | 10 | 0 |
> | w/o Reflector | 85 | 60 | 60 | 65 | 55 | 70 | 50 | 45 |
> | **MALLVi (Ours)** | **100** | **95** | **90** | **90** | **80** | **80** | **85** | **75** |
>
> *Table 1 : Success rates (%) on 8 real-world tasks with 20 repetitions.*
>
> **Why MALLVi performs better:**
>
> - Unlike VoxPoser, our multi-agent decomposition supports long-horizon reasoning and persistent memory.
> - Unlike ReKep, which is specialized for relational keypoints, our framework generalizes zero-shot across diverse task categories (reasoning, geometry, spatial sequencing).

---

### Author Response · Authors · 2025-11-28

The authors wish to express their sincere gratitude to the reviewers for their thorough review and helpful recommendations. We have carefully considered all comments and have revised the manuscript accordingly. To ensure transparency and facilitate the re-review process, every modification in the paper's text is marked with blue font color.

---

### Meta-Review · Area_Chair_jkRc · 2026-01-18

**Summary:**

The paper proposes MALLVi, a modular LLM and VLM-based framework for language-conditioned robotic manipulation. Tasks are decomposed into “atomic instructions” handled by specialized agents for planning, perception, grounding, execution, and verification, with a Reflector providing closed-loop feedback via retries. The framework is evaluated on real-world tabletop tasks, subsets of VIMABench, and RLBench.

Reviewers acknowledged that the system is clearly structured, practically motivated, and well engineered. At the same time, they raised substantial concerns regarding novelty, the characterization of the approach as a multi-agent system, and the depth and breadth of the experimental evaluation.

**Reviewer Concerns:**

Across the reviews, a central concern is limited novelty relative to prior closed-loop and multi-agent systems such as RePlan, RePlanVLM, and MALMM. Reviewers characterized the contribution as largely an arrangement of existing components and prompt-based orchestration, rather than a fundamentally new method. The rebuttal argues that agent-specific feedback routing, shared GraphState memory, and targeted retries distinguish MALLVi. However, these points primarily clarify design choices, and it remains unclear whether they constitute a substantive algorithmic advance beyond existing approaches.

Several reviewers also questioned whether the system constitutes a genuine multi-agent framework. The reviews emphasize that agents operate in a largely fixed execution order and communicate through shared state, without demonstrated dialogue, negotiation, or joint decision making. While the rebuttal clarifies internal mechanisms, the system still appears closer to a sequential pipeline of prompt-based modules than to an interactive multi-agent system.

There were also shared concerns about the experimental evaluation. Reviewers requested deeper module-level analysis, clearer reporting of low-level control, broader benchmark coverage, and reporting of inference cost and latency. Although the rebuttal adds baselines, clarifications, and additional results, these additions do not fully address concerns about evaluation breadth, task diversity, and generality.

**Reviewer Scores:**

Reviewer AM85 (score: 4) is unlikely to change their assessment substantially. Their main concerns around missing module-level analysis and limited baseline comparisons are only partially addressed.

Reviewer oLUG (score: 4) would likely remain near the original score. While some experimental clarifications were added, the reviewer emphasized evaluation breadth and fairness as the dominant factors in their assessment.

Reviewer JT5v (score: 2) is unlikely to raise their score, as their concerns focus on fundamental issues of novelty, multi-agent interaction, and missing inference cost analysis, which are not fully resolved.

Reviewer d7c7 (score: 2) is also unlikely to change their score. Their concerns regarding unclear novelty, insufficient task diversity, missing modern baselines, and oversimplified failure recovery reflect structural limitations of the current submission.

Overall, even accounting for the rebuttal, a significant upward shift in reviewer scores is unlikely.

---

### Decision · Program_Chairs · 2026-01-26

Reject